# Relation of urinary bisphenol concentration and diabetes or prediabetes in French adults: A cross-sectional study

**Julie Delepierre, Sandrine Fosse-Edorh, Clémence Fillol, Clara Piffaretti** *

Santé Publique France, The French National Public Health Agency, Saint Maurice, France

* clara.piffaretti@santepubliquefrance.fr

## Abstract

### Background

International research has recently shown an association between exposure to bisphenol A (BPA) and the risk of diabetes, although limited results are available for exposure to bisphenol S (BPS) and bisphenol F (BPF). The aim of this study was to examine the relationships between impregnation with BPA, BPS, and BPF and the prevalence of diabetes or prediabetes in the French adult population.

### Methods

Based on the Esteban cross-sectional study, 852 adults aged 18 to 74 years living in France were included. To assess the link between urinary concentration of BPA, BPS and BPF and a state of dysglycemia (diabetes or prediabetes), logistic regression multivariable models were performed and adjusted for known risk factors for diabetes and urine creatinine concentration.

### Results

The percentage of included individuals with diabetes or prediabetes was 17.8% (95% CI = [15.3–20.4]). Urinary BPA concentration was significantly higher in people with diabetes or prediabetes, independent of the known risk factors for diabetes (OR for an increase of 0.1 units in log-transformed concentration of BPA (µg/L) = 1.12; 95%CI = [1.05–1.19], p < 0.001). However, we did not find any significant independent association between urinary BPS and BPF levels and the prevalence of diabetes or prediabetes.

### Conclusions

In this sample, considering the diabetes risk factors, diabetes or prediabetes was positively associated with higher urinary BPA concentration but not with urinary BPS and BPF concentrations. However, analysis of prospective longitudinal studies are still necessary to demonstrate a causal link between bisphenol exposure and the risk of diabetes or prediabetes.

**Data Availability Statement:** Due to legal and ethical concerns, the data underlying the results presented in the study are available on justified request from Santé publique France, the French

national public health agency (https://www.
santepubliquefrance.fr/).

**Funding:** The French National Public Health Agency
runs the French human biomonitoring programme,
which is funded by the French Ministries of Health
and Environment. The funders were part of a
steering committee but had no role in data
collection and analysis, decision to publish, or
preparation of the manuscript.

**Competing interests:** The authors have declared
that no competing interests exist.

## Introduction

Type 2 diabetes is a chronic disease caused by the insufficient secretion of insulin by the pancreas or the reduced action of the insulin produced. The burden of diabetes-associated mortality and morbidity as well as the cost of the disease make it a major public health challenge. The prevalence of diabetes among adults has increased worldwide from 4.7% in 1980 to 8.5% in 2014 according to the World Health Organization (WHO) [1]. It was estimated to be 10.5% in 2021, rising to 12.2% in 2045, according to the International Diabetes Federation [2]. Age, sex, obesity, sedentary lifestyle, and family history are known risk factors for type 2 diabetes, but the unfavorable development of the disease has prompted the search for new factors likely to induce dysglycemia [3, 4]. Prediabetes is a high-risk state of diabetes and cardiovascular diseases, but effective strategies implemented at this stage can delay or prevent the development of diabetes [5]. In France, according to data from the French Nutrition and Health Survey (ENNS) in 2006–2007, the prevalence of diabetes and prediabetes was estimated at 5.1% and 5.6%, respectively, while it reached 7.4% and 9.9% in the Esteban survey in 2014–2016 [6].

Recently, international research on the potential contribution of exposure to environmental chemicals in the etiology of diabetes has rapidly expanded. Although causality has not yet been demonstrated, an overall positive association has been found between certain chemical substances present in the environment and diabetes. Therefore, it is necessary to continue research to improve our understanding of the role played by environmental exposures and facilitate the implementation of prevention strategies [7].

Certain environmental chemicals are endocrine-disrupting chemicals, that is, substances capable of interfering with the hormones present in the blood and acting on the organs regulating blood sugar and lipemia. Observational studies have shown a positive relationship between endocrine-disrupting chemicals and the diabetes epidemic [8–11]. Bisphenol A (BPA) is a known endocrine-disrupting chemical, meaning that it may be a risk factor for diabetes. The ubiquity of BPA in the environment, especially in food packaging, drinking water, dental materials, thermal paper, household dust, and tobacco smoke, makes its urinary concentration detectable in more than 90% of individuals [8–11]. Experimental studies have shown that BPA plays a role in the development of insulin resistance, adipogenesis, and dysfunction of β cells in the pancreas [12, 13]. In mice, research showed that long-term BPA exposure resulted in increased adipose tissue mass and hyperglycemia [12]. In humans, according to two meta-analyses mainly based on cross-sectional studies, urinary BPA concentrations are significantly associated with the risk of diabetes [14, 15]. To our knowledge, only one study has examined the relationship between prediabetes and BPA: the authors observed a positive independent association between higher levels of urinary BPA and prediabetes [16].

Little research has been done on other types of bisphenols such as bisphenol S (BPS) and bisphenol F (BPF). However, a case-control study conducted in China showed a significant association between urinary BPS and diabetes mellitus [17]. In France, a prospective study undertaken in a population sample between 1994 and 1996 reported that the detection of the main metabolite of BPS in urine was associated with incident diabetes [18]. It would therefore be interesting to confirm these results with more recent data.

Despite the description of these biological mechanisms and the associations found between bisphenols and diabetes at the international level, few studies have been conducted in France. Furthermore, the French national health study on environment, biomonitoring, physical activity, and nutrition known as *Esteban* [19] constitutes a relevant data source to study this association. Therefore, the objective of this study is to examine the relationship between exposure to BPA, BPS, and BPF and the prevalence of diabetes or prediabetes in the French adult population based on the Esteban study.

## Methods

### Esteban study

Esteban is a cross-sectional study that was carried out in the general population of mainland France between April 2014 and March 2016. A national sample of children aged 6 to 17 and adults aged 18 to 74 was included. Inclusions were balanced according to the seasonality of environmental and food exposures. The sampling plan was probabilistic at three degrees (primary units, households, individual), with the stratification on the region and the degree of urbanization. Data were collected from interview-guided questionnaires, self-questionnaires, a 24-hour dietary recall, and a medical examination with biological samples (urine, blood, and hair) and clinical data collection (anthropometric measurements, blood pressure, etc.). So as not to disturb the biological analyses, participants had to be fasting, and they were not allowed to smoke in the previous 2 hours. Data from the study were matched with individual data from the National Health Data System (SNDS). Bisphenol levels were measured in a subsample selected by random draw among participants who accepted to participate in the Esteban survey and had at least one spot urine sample. Approval for the study was obtained from the French data protection authority and a bioethics committee. A written informed consent form for participation in the study was signed in triplicate by all the participants [19].

Detailed information about the Esteban protocol and the characteristics of participants has been previously published [19].

### Study population

To study the relationship between bisphenol exposure and the risk of diabetes or prediabetes, Esteban participants had to be at least 18 years. Exclusion criteria were: absence of at least one questionnaire (interview-guided and self-administered), absence of medical examination, having gestational diabetes, absence of fasting serum glucose measurement and absence of measurement of urinary concentration of BPA, BPS, and BPF.

### Diabetes and prediabetes

The different stages of dysglycemia (prediabetes, diagnosed and undiagnosed diabetes) were defined according to the answers to questionnaires, diabetes medication reimbursements recorded in the SNDS, and fasting serum glucose measurement. Serum glucose was measured in a blood sample using a tube containing glycolysis inhibitors [19]. Among people who did not report diabetes or receive diabetes medication (codes beginning with A10 in the Anatomical Therapeutic Chemical Classification System), undiagnosed diabetes was defined by a serum glucose level of at least 7.0 mmol/L and prediabetes by a value between 6.1 and 7.0 mmol/L, as defined by WHO [6].

### Bisphenol exposure

Exposure to BPA, BPS, and BPF was assessed by the analytical measurement of biomarkers in urine, as the urinary route is the primary route of excretion in humans [20].

The first morning urine was collected by participants at home in a polypropylene container.

The urinary concentrations of total bisphenols were quantified by gas chromatography coupled to a tandem mass spectrometry. The limit of detection (LOD) and the limit of quantification (LOQ) were, respectively, 0.01 μg/L and 0.09 μg/L for BPA, 0.003 μg/L and 0.006 μg/L for BPS, and 0.01 μg/L and 0.02 μg/L for BPF. The three bisphenols were detected and quantified in 100% of the samples analyzed [21].

To account for urinary dilution, the urinary creatinine concentration was taken into account and measured using the kinetic Jaffe method [22].

Bisphenol concentrations measured in urine corresponded to a recent exposure due to the short half-life of these compounds (less than 10 hours after oral exposure).

## Other factors

Participants' sociodemographic information (age, sex, education level) and duration of smoking were collected in the interview-guided questionnaires. Body mass index (BMI) was calculated from weight and height measured during the medical examination. High blood pressure was defined on the basis of medical examination data (systolic blood pressure $\geq$ 140 mmHg or diastolic blood pressure $\geq$ 90 mmHg) or the SNDS with at least one reimbursement of antihypertensive treatment in the year preceding the medical examination [23]. Hypercholesterolemia was defined on the basis of medical examination data (LDL cholesterol level > 1.6 g/L) or the SNDS with at least one reimbursement of lipid-lowering treatment in the year preceding the medical examination. The level of sedentariness was defined by the daily duration of sedentary activities, that is, corresponding to less than 1.6 MET (Metabolic Equivalent Task) [24]. A low level of sedentariness corresponds to a duration of less than 3 hours per day, a moderate level to between 3 and 7 hours per day, and a high level to more than 7 hours per day [25]. Energy intake was measured from dietary data collected using three 24-hour recalls. The participants were not informed in advance about the recall day to limit changes to their eating habits [19].

## Statistical analyses

Numbers and percentages were described for qualitative variables. Mean and standard deviation or median as well as 10th and 90th percentiles were described for quantitative variables. When a variable had more than 5% missing data, a class of missing values was added.

To compare people with diabetes or prediabetes and those without, bivariate analysis was performed after verifying the validity conditions. Means for the quantitative variables were compared using the Student test for independent samples. Percentages for the categorical variables was compared using Pearson's chi-squared test.

To limit their dispersion as well as their skewed distribution, the bisphenol concentration values were log-transformed (with base 10). The relationships between the three bisphenols and diabetes or prediabetes were explored by fractional polynomials to determine if they should be considered continuously or not.

The adjusted odds ratio (OR) and 95% confidence interval (CI) were calculated using logistic regression multivariable models to explain diabetes or prediabetes as a function of the log-transformed urinary concentration in BPA, BPS, and BPF separately. The models were adjusted to known risk factors for diabetes based on the literature and significantly associated in bivariate analysis: age, sex, education level (less than high school graduate vs high school graduate or more), BMI (normal: BMI < 25 kg/m$^2$, overweight: BMI between 25 and 30 kg/m$^2$, and obese: BMI > 30 kg/m$^2$), hypertension, hypercholesterolemia, level of sedentariness (low, moderate, or high), duration of smoking (never smoker, smoker for less than 20 years, or smoker for more than 20 years), and daily energy intake (including alcohol intake in kilocalories per day in tertiles). Urinary creatinine concentrations were introduced into the models after logarithmic transformation. Finally, the validity conditions and model suitability were confirmed.

Statistical analyses were performed with SAS Enterprise Guide 7.1 and Stata/SE 15.0 software. Statistical tests with a p-value $\leq$ 0.05 were considered significant.

## Results

Between 2014 and 2016, 3,476 adults were recruited in the Esteban study. The "Esteban population" comprised 2,345 participants after excluding individuals for one of the following reasons: absence of medical examination (N = 973), absence of at least one questionnaire (N = 60), absence of fasting serum glucose measurement (N = 97), or gestational diabetes during the study (N = 1).

Among the remaining 2,345 participants, the urinary concentrations of bisphenols were measured in 852 participants ("bisphenol population"). The S1 Table describes the two populations, which present similar characteristics. BPA, BPS, and BPF levels in the bisphenol population are shown in Fig 1. BPA levels are higher in participants with diabetes or prediabetes compared to people without diabetes or prediabetes, while BPS and BPF levels are similar across the two groups.

In bivariate analyses (Table 1), compared to people without diabetes or prediabetes, people with diabetes or prediabetes were significantly older with a significantly higher proportion of men and lower level of education. They were also significantly more overweight and obese, more likely to be long-term smokers, and had higher blood pressure and hypercholesterolemia.

The relationships between the three bisphenols and diabetes or prediabetes were explored by fractional polynomials. Based on the results of this analysis (*results not shown*), urinary bisphenol concentrations were considered continuously.

The associations between bisphenol urinary concentrations and diabetes or prediabetes are shown in Table 2.

In the crude models, the OR for an increase of 0.1 units in log-transformed urinary concentration of BPA (μg/L) was 1.08 (95% CI = [1.03–1.13]). After adjusting for covariates, the association of BPA with diabetes or prediabetes remained significant, with an adjusted OR of 1.12 (95% CI = [1.05–1.19]). However, in the crude and adjusted models, the associations of BPS and BPF with diabetes or prediabetes were not significant.

## Discussion

In this cross-sectional study conducted on a sample of the French general population, we found that urinary BPA concentrations were significantly higher in people with diabetes or prediabetes, regardless of the known risk factors for diabetes. In addition, we did not find any significant independent association between urinary BPS and BPF levels and diabetic or prediabetic status.

International studies show a high methodological variability in their evaluation of exposure (urinary concentration of total or free bisphenol, metabolite concentration, correction or not for urinary creatinine levels) as well as outcomes (self-reported diabetes, diabetes medication, fasting blood glucose, glycated hemoglobin). As exposure to BPA is suspected of causing hyperglycemia, we chose to study the association between this exposure and diabetic or prediabetic status.

Several studies [26–29] and meta-analyses [14, 15] have observed significant associations between BPA exposure and the risk of diabetes. For example, the American NHANES cross-sectional study (National Health and Nutrition Examination Survey, 2003–2008) showed that urinary BPA concentrations were positively associated with the prevalence of type 2 diabetes in adults aged 20 years and over [27]. In France, to our knowledge, few studies have explored the link between diabetes or prediabetes and bisphenol exposure. A case-cohort study was performed as part of the French prospective cohort D.E.S.I.R. (Data from an Epidemiological Study on the Insulin Resistance Syndrome) [18] in people aged 30 to 65 years (enrollment in

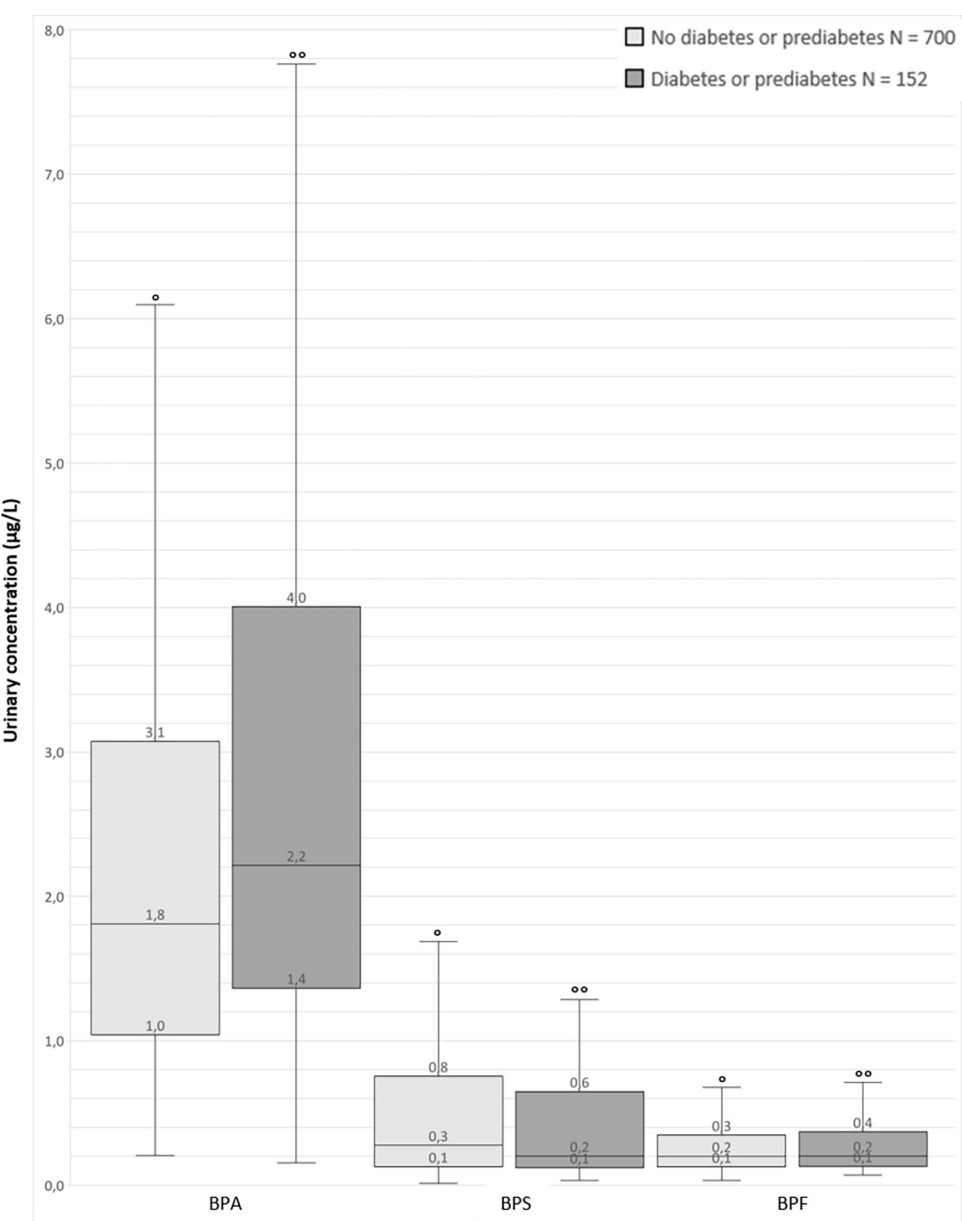

**Fig 1. Description of urinary concentrations of bisphenols (μg/L) in the bisphenol population by boxplots.** Within each box, horizontal black line denotes median value; boxes extend from the 1st to the 3rd quartile (= inter-quartile range IQR) of each group's distribution of value. The ends of the extended vertical lines indicate the 1st quartile minus 1.5 times the IQR and the 3rd quartile plus 1.5 times the IQR; observations outside this range are not represented (BPA ˚ 59 observations >6.1˚˚ 11 observations >7.8 / BPS ˚ 80 observations >1.7˚˚ 22 observations >1.3 BPF ˚ 52 observations > 0.7˚˚ 15 observations > 0.7; no extreme low values were observed). BPA: bisphenol A, BPS: bisphenol S, BPF: bisphenol F.

1994–1996 and follow-up up to 9 years). This study showed that compared to participants with average BPA glucuronide concentrations below the first quartile, participants in the second, third, and fourth quartiles of exposure had an increased risk of type 2 diabetes (Hazard Ratio (HR) = 2.56 [95% CI: 1.16–5.65], 2.35 [95% CI: 1.07–5.15], and 1.56 [95% CI: 0.68–3.55], respectively). In addition, the detection (≥ LOD) of BPS glucuronide in urine at baseline, after 3 years, or at both time points was associated with incident diabetes (HR = 2.81 [95% CI:

**Table 1. Characteristics of participants according to diabetic or prediabetic status.**

| | No diabetes, no prediabetes N = 700 | Diabetes or prediabetes N = 152 | p-value |
|---|---|---|---|
| **Age** in years (+/-SD) | 50.2 (± 12.8) | 58.0 (± 11.4) | < 0.0001 |
| **Sex** | | | < 0.0001 |
| Men | 285 (40.7%) | 100 (65.8%) | |
| Women | 415 (59.3%) | 52 (34.2%) | |
| **Education level** | | | < 0.001 |
| Less than high school | 174 (24.9%) | 58 (38.2%) | |
| High school graduate or more | 526 (75.1%) | 94 (61.8%) | |
| **Body mass index** | | | < 0.0001 |
| Normal: < 25 kg/m$^2$ | 415 (59.4%) | 29 (19.2%) | |
| Overweight: 25–30 kg/m$^2$ | 207 (29.6%) | 69 (45.7%) | |
| Obesity: > 30 kg/m$^2$ | 77 (11.0%) | 53 (35.1%) | |
| **High blood pressure** | | | < 0.0001 |
| Yes | 183 (26.1%) | 94 (61.8%) | |
| No | 443 (63.3%) | 44 (28.9%) | |
| Missing | 74 (10.6%) | 14 (9.2%) | |
| **Hypercholesterolemia** | | | 0.004 |
| Yes | 181 (25.9%) | 59 (38.8%) | |
| No | 417 (59.6%) | 71 (46.7%) | |
| Missing | 102 (14.6%) | 22 (14.5%) | |
| **Sedentariness** | | | 0.094 |
| Low | 91 (13.5%) | 11 (7.5%) | |
| Moderate | 311 (46.3%) | 78 (53.1%) | |
| High | 270 (40.2%) | 58 (39.5%) | |
| **Duration of smoking** | | | < 0.001 |
| Never smoker | 372 (53.9%) | 62 (41.1%) | |
| Less than 20 years | 213 (30.9%) | 45 (29.8%) | |
| More than 20 years | 105 (15.2%) | 44 (29.1%) | |
| **Energy intake** | | | 0.764 |
| < 1664 kcal/day | 220 (32.4%) | 50 (33.8%) | |
| 1664-2149kcal/day | 228 (33.5%) | 45 (30.4%) | |
| > 2149 kcal/day | 232 (34.1%) | 53 (35.8%) | |

Results are presented as mean (± standard deviation) for continuous variables and as number (percentage) for categorical variables. Means of age were compared using the Student test for independent samples. Percentages were compared using Pearson's chi-squared test.

**Table 2. Odds ratios (ORs) [95% confidence interval (CI)] of diabetes or prediabetes for an increase of 0.1 units in log-transformed urinary concentrations of bisphenols (µg/L) in logistic regression analyses.**

| | Unadjusted (N = 852) | | Adjusted[†] (N = 806) | |
|---|---|---|---|---|
| | OR [95% CI] | p-value | OR (95% CI) | p-value |
| **BPA total[‡]** | 1.08 [1.03–1.13] | < 0.001 | 1.12 [1.05–1.19] | < 0.001 |
| **BPS total[‡]** | 0.99 [0.96–1.02] | 0.53 | 0.99 [0.96–1.02] | 0.44 |
| **BPF total[‡]** | 1.03 [0.98–1.08] | 0.25 | 1.04 [0.98–1.11] | 0.20 |

[†]Adjusted for age, sex, education level, body mass index, high blood pressure, hypercholesterolemia, level of sedentariness, duration of smoking, daily energy intake (including alcohol intake), and urinary creatinine concentration

[‡]BPA: log-transformed bisphenol A, BPS: log-transformed bisphenol S, BPF: log-transformed bisphenol F

1.74–4.53]). The results of our study on BPA are consistent with the results of the D.E.S.I.R. study, but in a population with a wider age group and more recent years, and with the inclusion of the notion of prediabetes in our analyses.

Our study has several strengths. First, diabetes cases were identified from different data sources (self-reporting, SNDS, and blood samples) to limit the biases associated with self-reporting and identify cases of undiagnosed diabetes and prediabetes, even though we only had access to one glycemic dosage. The absence of a second blood glucose test or an HbA1c test may have influenced certain results, in particular for prediabetes. However, we were able to base the diagnosis of diabetes also on healthcare reimbursement data and on a self-administered questionnaire. In addition, to ensure the quality of the data, the procedures were standardized and the biological samples subjected to quality controls. Finally, to limit the confounding bias that may arise in the relationship between exposure to bisphenols and risk of diabetes or prediabetes, the multivariable models were adjusted to the majority of known risk factors for diabetes. The models were also adjusted to dietary intake, which is a major potential confounding factor for the relationship between BPA and diabetes that is rarely taken into account in research [14]. However, daily energy intake does not fully represent dietary intake.

This study has also several limitations. First, family history of diabetes was unavailable and thus not considered in the multivariable models. However, it is a known and important risk factor for diabetes, which could potentially modify the response of the organism when exposed to bisphenols or influence the behavior of these individuals and thus their bisphenol exposure.

Exposure to bisphenols was assessed using a single measure of their urinary concentration, which may not reflect chronic exposure because of the short biological half-life of bisphenols. In addition, due to the intraindividual variability in the biological concentrations of biomarkers over the course of a day or week, the risk of error in the individual estimation of exposure to pollutants and the classification of unexposed people should not be excluded. However, to limit this variability, the first morning urine specimens were collected.

Our study highlights an association between exposure to BPA and dysglycemia status (diabetes or prediabetes). However, due to the cross-sectional nature of the study, longitudinal studies including repeated measurements of exposures to bisphenol and the occurrence of diabetes or prediabetes are necessary to highlight an association with the development of diabetes. However, research is often forced to use cross-sectional studies given the complexity and cost associated with longitudinal studies [26, 27, 30].

In addition, simultaneous exposure to several environmental pollutants could be a confounding factor in the relationship between diabetes or prediabetes and urinary BPA levels. However, given the complexity of assessing co-exposure, this would require a larger number of participants. Furthermore, studies that control exposure to other types of endocrine disruptors [31, 32] have not been able to demonstrate that other pollutants change the associations between exposure to BPA and diabetes.

## Conclusion

In our sample of the metropolitan French adult population, when taking into account the main risk factors for diabetes, we observed that the diabetic or prediabetic status was positively associated with urinary BPA concentration. We did not highlight an association with urinary concentrations of BPS and BPF.

Since BPA is considered an endocrine-disrupting chemical with harmful health effects, its use is increasingly regulated. Since 2015, the ban on BPA in materials that come into contact with food has been applied in France. Measures have also been taken to restrict its use in thermal papers and cosmetic products. In light of the results of our study, this regulation of BPA

seems justified. However, following the ban on BPA, the use of other bisphenols such as BPS and BPF has increased. As our study is based on data collected between 2014 and 2016, it deserves to be renewed with more recent data to explore the association with BPS and BPF.

In addition, as diabetes is a major public health concern and environmental pollutants may be involved in the etiology of diabetes, analysis of prospective longitudinal studies are still needed to demonstrate a causal link between exposure to bisphenols and the onset of diabetes or prediabetes.

## Supporting information

**S1 Table. Characteristics of the two populations.** Results are presented as mean (± standard error) for continuous variables and as number (percentage) for categorical variables. (DOCX)

## Acknowledgments

The authors would like to thank the Centers for Health Examinations, the Cetaf and the laboratories involved in the collection and the analysis of samples, as well as the entire Esteban project team and the survey participants.

## Author Contributions

**Conceptualization:** Sandrine Fosse-Edorh, Clémence Fillol, Clara Piffaretti.

**Data curation:** Julie Delepierre.

**Formal analysis:** Julie Delepierre.

**Methodology:** Julie Delepierre, Sandrine Fosse-Edorh, Clémence Fillol, Clara Piffaretti.

**Supervision:** Sandrine Fosse-Edorh, Clémence Fillol, Clara Piffaretti.

**Writing – original draft:** Julie Delepierre, Clara Piffaretti.

**Writing – review & editing:** Sandrine Fosse-Edorh, Clémence Fillol, Clara Piffaretti.

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
