## [Decision Letter · Decision Letter 0]

5 Sep 2022

PONE-D-22-22640Relation of urinary bisphenol concentration to risk of diabetes or prediabetes in French adults: a cross-sectional studyPLOS ONE

Dear Dr. Piffaretti,

Thank you for submitting your manuscript to PLOS ONE. After careful consideration, we feel that it has merit but does not fully meet PLOS ONE’s publication criteria as it currently stands. Therefore, we invite you to submit a revised version of the manuscript that addresses the points raised during the review process.

Dear Author, there is a possibility that your article may be accepted for publication. However, there are some issues that need to be clarified. In this way, they follow the estimates for appreciation.

We look forward to receiving your revised manuscript.

Kind regards,

Ana Claudia Morais Godoy Figueiredo, Ph.D

Academic Editor

PLOS ONE

2. Please ensure that you have specified (1) whether consent was informed and (2) what type you obtained (for instance, written or verbal, and if verbal, how it was documented and witnessed). If your study included minors, state whether you obtained consent from parents or guardians. If the need for consent was waived by the ethics committee, please include this information.

Additional Editor Comments :

Dear Author,

It is necessary to inform whether a normality test was applied to identify the type of parametric and non-parametric test. What was the criterion for choosing the student's t test?

Additionally, although it is a population-based study, it is important to show that the sample used for the hypothesis tested has sufficient power. In this way, calculate the power of your results against the sample used and present it at the end of the discussion. You already inform that the power is insufficient in the end, but how much would it be? And how this undermines the scientific evidence produced.

Is the evidence valid for decision-making by health professionals? Talk more about this aspect in the discussion.

Reviewers' comments:

Reviewer's Responses to Questions

**Comments to the Author**

1. Is the manuscript technically sound, and do the data support the conclusions?

Reviewer #1: Partly

Reviewer #2: Yes

2. Has the statistical analysis been performed appropriately and rigorously? 

Reviewer #1: Yes

Reviewer #2: Yes

3. Have the authors made all data underlying the findings in their manuscript fully available?

Reviewer #1: No

Reviewer #2: Yes

4. Is the manuscript presented in an intelligible fashion and written in standard English?

Reviewer #1: Yes

Reviewer #2: Yes

5. Review Comments to the Author

Reviewer #1: It is not clear whether the data is available, as required by PLOSOne `The PLOS Data policy requires authors to make all data underlying the findings described in their manuscript fully available without restriction, with rare exception`

I have responded `partly` to the question `Is the manuscript technically sound, and do the data support the conclusions?` as I am not sure of some of the statistical analyses, and I expect the authors to respond to my comments in the attached review document. I indicate above that the statistics have been performed correctly.

Reviewer #2: Congratulations to the authors for conducting the study and reporting the results in this manuscript.

The manuscript is concise and of pleasant to read.

Some contributions:

1. Describe in the method the exclusion criteria of the participants - this information was presented only in the results.

2. The results presented in Table 1 do not contribute to answer the research question. I suggest removing these results and going straight to the results presented in Table 2.

3. As it is a cross-sectional study, it is better to refer to the odds ratio and not the risk.

4. The first paragraph of the conclusion is general and does not add information to the reader. I suggest starting with the text of the second paragraph.

6. PLOS authors have the option to publish the peer review history of their article (what does this mean?). If published, this will include your full peer review and any attached files.

Reviewer #1: No

Reviewer #2: No

---

## [Author Response · Author response to Decision Letter 0]

25 Oct 2022

Response: It has been done, and the files have been renamed.

2. Please ensure that you have specified (1) whether consent was informed and (2) what type you obtained (for instance, written or verbal, and if verbal, how it was documented and witnessed). If your study included minors, state whether you obtained consent from parents or guardians. If the need for consent was waived by the ethics committee, please include this information.

Response: Previously, it was only specified in the Esteban protocol article (reference 19). We added this sentence directly in the “Esteban study” part of the Methods: “An informed consent form for participation in the study was signed in triplicate by the participants.”

Response: The authors received no specific funding for this work, and no grants have been received. However, the French national public health agency, which is the investigator of the study, was funded by the public authorities. We added this sentence in the cover letter “The French National Public Health Agency runs the French human biomonitoring programme, which is funded by the French Ministries of Health and Environment.” We respectfully hope that these clarifications have answered all your questions.

Reponse: The data is public but cannot be shared on the internet (open data) for ethical and legal reasons, imposed by the French regulatory authorities. That is why we specified "Due to legal and ethical concerns, the data underlying the results presented in the study are available, on justified request, from Santé publique France, the French national public health agency. (https://www.santepubliquefrance.fr/)" Santé publique France is a public government agency that can be contacted via its website https://www.santepubliquefrance.fr/.

Additional Editor Comments :

Dear Author,

It is necessary to inform whether a normality test was applied to identify the type of parametric and non-parametric test. What was the criterion for choosing the student's t test?

Response: Dear Editor, thank you for your detailed comments and vigilance. Only the Student test was used. This was rectified in the methods and in the note of Table 2 [table 1 in the revised manuscript] (as mentioned by reviewer #1, we only describe one continuous variable in Table 2[table 1 in the revised manuscript]).

The normality test was applied to identify the statistic test to use. We also looked at the graphical aspect of the data.

Additionally, although it is a population-based study, it is important to show that the sample used for the hypothesis tested has sufficient power. In this way, calculate the power of your results against the sample used and present it at the end of the discussion. You already inform that the power is insufficient in the end, but how much would it be? And how this undermines the scientific evidence produced.

Response: Thank you for this interesting comment. In light of your contribution and also to comply with Reviewer#1’s comment number 27 (i.e. “line 288: insufficient statistical power is not an argument, as your ORs are far from being statistically significant”), we propose to remove the following sentence from the discussion. Indeed, it added confusion and was not entirely correct, as stated by Reviewer#1. We have deleted the following sentence : “Finally, although the sample size (N = 852) is reasonable compared to other international studies, the absence of a significant association between urinary BPS and BPF concentrations and the diabetic or prediabetic status may be due to an insufficient statistical power.”

Is the evidence valid for decision-making by health professionals? Talk more about this aspect in the discussion.

Response: Our evidence is valid and deserves to be transmitted to decision-makers. Although, it concerns more government decisions than day-to-day clinical decision of health professionals. We said in our paper in the conclusion that: “In light of the results of our study, this regulation of BPA seems justified”. However, the most useful studies for health policy makers are those that establish causality. We are convinced of the value of the information we are providing with this cross-sectional study, but we have nevertheless tempered the interpretation of our results, since the study design only allows us to generate or provide elements of confirmation of certain hypotheses. We have specified that: “prospective longitudinal studies are necessary needed to demonstrate a causal link between exposure to bisphenols and the onset of diabetes or prediabetes”.

Review Comments to the Author

Reviewer #1: It is not clear whether the data is available, as required by PLOSOne `The PLOS Data policy requires authors to make all data underlying the findings described in their manuscript fully available without restriction, with rare exception`

Response: We answered the Editor directly. The data is public but can only be available on demand, due to ethics and legal reasons.

I have responded `partly` to the question `Is the manuscript technically sound, and do the data support the conclusions?` as I am not sure of some of the statistical analyses, and I expect the authors to respond to my comments in the attached review document. I indicate above that the statistics have been performed correctly.

Response: We answered every comment in the attached document (“Response to reviewers”).

Reviewer #2: Congratulations to the authors for conducting the study and reporting the results in this manuscript.

The manuscript is concise and of pleasant to read.

Some contributions:

1. Describe in the method the exclusion criteria of the participants - this information was presented only in the results.

Response: Thank you for the comment. We clarified the exclusion criteria in the study population paragraph in the method. The new paragraph is as follows:

“To study the relationship between bisphenol exposure and the risk of diabetes or prediabetes, Esteban participants had to be at least 18 years. Exclusion criteria were: absence of at least one questionnaire (interview-guided and self-administered), absence of medical examination, having gestational diabetes, absence of fasting serum glucose measurement and absence of measurement of urinary concentration of BPA, BPS, and BPF.”

2. The results presented in Table 1 do not contribute to answer the research question. I suggest removing these results and going straight to the results presented in Table 2.

Response: Table 1 describes the total populations of the Esteban study (n=2345) and the population of our study (n=852). I agree that it is not essential, however it does support that the two population present similar characteristics. We propose to remove it from the main manuscript but add it in a Supporting information section.

3. As it is a cross-sectional study, it is better to refer to the odds ratio and not the risk.

Response: Thank you for this point. We agree and thought we tried to make this clear in the first draft of the manuscript. We have re-read the manuscript with this particular point in mind. We don’t use the word risk anymore and have made changes throughout the manuscript.

4. The first paragraph of the conclusion is general and does not add information to the reader. I suggest starting with the text of the second paragraph.

Response: Thank you, we agree. We started the conclusion with the second paragraph, and enrich the last paragraph with some elements of the first paragraph that was deleted.

The last paragraph became: “In addition, as diabetes is a major public health concern and environmental pollutants may be involved in the etiology of diabetes, prospective longitudinal studies are still needed to demonstrate a causal link between exposure to bisphenols and the onset of diabetes or prediabetes.”

---

## [Decision Letter · Decision Letter 1]

19 Dec 2022

PONE-D-22-22640R1Relation of urinary bisphenol concentration and diabetes or prediabetes in French adults: a cross-sectional studyPLOS ONE

Dear Dr. Piffaretti,

Thank you for submitting a revised copy of the above referenced manuscript to PLOS ONE. As shown at the end of this email, **Reviewer # 1** still has some minor concerns/comments that need to be addressed by the authors before a final decision is made. Therefore, we invite you to submit a revised version of the manuscript that addresses the points raised during the review process.

We look forward to receiving your revised manuscript.

Kind regards,

Iman Al-Saleh

Academic Editor

PLOS ONE

Journal Requirements:

**Reviewers' comments**:

Reviewer's Responses to Questions

**Comments to the Author**

1. If the authors have adequately addressed your comments raised in a previous round of review and you feel that this manuscript is now acceptable for publication, you may indicate that here to bypass the “Comments to the Author” section, enter your conflict of interest statement in the “Confidential to Editor” section, and submit your "Accept" recommendation.

Reviewer #1: (No Response)

**Reviewer #2: **All comments have been addressed

2. Is the manuscript technically sound, and do the data support the conclusions?

Reviewer #1: Yes

Reviewer #2: Yes

3. Has the statistical analysis been performed appropriately and rigorously? 

Reviewer #1: Yes

Reviewer #2: Yes

4. Have the authors made all data underlying the findings in their manuscript fully available?

Reviewer #1: Yes

Reviewer #2: Yes

5. Is the manuscript presented in an intelligible fashion and written in standard English?

Reviewer #1: Yes

Reviewer #2: Yes

6. Review Comments to the Author

**Reviewer #1**: Thank you for your responses to my comments. The paper is easy to read, well documented.

I still have some comments on the revised manuscript.

1. Line 16 – it should be `prevalence` not `risk`

2. Line 20 and elsewhere in the manuscript, you should write `multivariable models` not `multivariate models`.

3. Line 22 you quote a `percentage` not a `proportion`

4. line 24 and elsewhere in the text line 211, Table 2 legend) – this should be?:

OR for an increase of 0.1 units in log transformed concentration of BPA (�g/L)

5. `prevalence` not `risk`

6. Line 79 add `BPS, and BPF and the prevalence of diabetes or prediabetes…`

7. Line 147 – you correctly use the `standard deviation` not the `standard error`. Change this.

8. Table 1 the footnote should read `standard deviation` not `standard error`

9. Line 266 - `a French population` - it is not the entire French population

Reviewer #2: (No Response)

7. PLOS authors have the option to publish the peer review history of their article (what does this mean?). If published, this will include your full peer review and any attached files.

Reviewer #1: No

Reviewer #2: **Yes: **Priscilla Perez da Silva Pereira

---

## [Author Response · Author response to Decision Letter 1]

7 Mar 2023

Response: It has been done.

Review Comments to the Author

Reviewer #1: Thank you for your responses to my comments. The paper is easy to read, well documented.

I still have some comments on the revised manuscript.

1. Line 16 – it should be `prevalence` not `risk`

Thank you for the comment. We replaced ‘risk’ by ‘prevalence’.

2. Line 20 and elsewhere in the manuscript, you should write `multivariable models` not `multivariate models`.

Response: Thank you for this point. We changed ‘multivariate models’ to ‘multivariable models’ in lines 20, 160, 261 and 266.

3. Line 22 you quote a `percentage` not a `proportion`

Response: We replaced the term ‘proportion’ with ‘percentage’.

4. line 24 and elsewhere in the text line 211, Table 2 legend) – this should be?: OR for an increase of 0.1 units in log transformed concentration of BPA (µg/L)

Response: Thank you for this precision, the wording has been changed.

5. `prevalence` not `risk`

 Response: On line 28, we changed ‘risk’ to ‘prevalence’.

6. Line 79 add `BPS, and BPF and the prevalence of diabetes or prediabetes…`

Response: Thank you, we added ‘the prevalence of’ in the text.

7. Line 147 – you correctly use the `standard deviation` not the `standard error`. Change this.

8. Table 1 the footnote should read `standard deviation` not `standard error`

Response: Sorry for this mistake, we replaced ‘standard error’ by ‘standard deviation’ in the text and in the footnote of Table 1.

9. Line 266 - `a French population` - it is not the entire French population

Response: On line 227, we changed ‘of the French general population’ to ‘conducted on a sample of the French general population’. On line 288, we replaced ‘In our sample of the French adult population’ by ‘In our sample of the metropolitan French adult population’, the overseas territories not being included.

---

## [Editor Report · Decision Letter 2]

8 Mar 2023

Relation of urinary bisphenol concentration and diabetes or prediabetes in French adults: a cross-sectional study

PONE-D-22-22640R2

Dear Dr. Piffaretti,

We’re pleased to inform you that your manuscript has been judged scientifically suitable for publication and will be formally accepted for publication once it meets all outstanding technical requirements.

Kind regards,

Iman Al-Saleh

Academic Editor

PLOS ONE

---

## [Editor Report · Acceptance letter]

22 Mar 2023

PONE-D-22-22640R2 

Relation of urinary bisphenol concentration and diabetes or prediabetes in French adults: a cross-sectional study 

Dear Dr. Piffaretti:

I'm pleased to inform you that your manuscript has been deemed suitable for publication in PLOS ONE. Congratulations! Your manuscript is now with our production department. 

Kind regards, 

on behalf of

Dr. Iman Al-Saleh 

Academic Editor

PLOS ONE